# Altered Processing of Visual Food Stimuli in Adolescents with Loss of Control Eating

**DOI:** 10.3390/nu11020210

**Published:** 2019-01-22

**Authors:** Stefanie C. Biehl, Ulrich Ansorge, Eva Naumann, Jennifer Svaldi

**Affiliations:** 1Department of Clinical Psychology and Psychotherapy, University of Tuebingen, Schleichstrasse 4, 72076 Tuebingen, Germany; eva.naumann@mssm.edu (E.N.); jennifer.svaldi@uni-tuebingen.de (J.S.); 2Department of Clinical Psychology and Psychotherapy, University of Regensburg, Universitaetsstrasse 31, 93053 Regensburg, Germany; 3Faculty of Psychology, University of Vienna, Liebiggasse 5, 1010 Vienna, Austria; ulrich.ansorge@univie.ac.at; 4Eating and Weight Disorders Program, Icahn School of Medicine at Mount Sinai, One Gustave L. Levy Place, New York, NY 10029-6574, USA

**Keywords:** loss of control eating, adolescents, event-related potentials, P3, Go/NoGo task, visual food stimuli

## Abstract

Loss of control eating (LOC) constitutes a common eating pathology in childhood and adolescence. Models developed for adult patients stress a biased processing of food-related stimuli as an important maintaining factor. To our knowledge, however, no EEG study to date investigated the processing of visual food stimuli in children or adolescents with LOC. Adolescents with at least one self-reported episode of LOC in the last four weeks and a matched control group completed a modified Go/NoGo task, with a numerical target or non-target stimulus being presented on one side of the screen and an irrelevant high-calorie food or neutral stimulus being presented on the opposite side. Mean P3 amplitudes were analyzed. In Go trials, the LOC group’s mean P3 amplitudes were comparable irrespective of distractor category, while for NoGo trials, mean P3 amplitudes were significantly higher when the distractor was a high-calorie food stimulus. This pattern was reversed in the control group. Results are interpreted in light of Gray’s reinforcement sensitivity theory. They might reflect altered processes of behavioral inhibition in adolescents with LOC upon confrontation with visual food stimuli.

## 1. Introduction

With a prevalence of 1.6–2.0% [1,2,3], loss of control eating (LOC) constitutes the most common eating pathology in childhood and adolescence. The troubling experience of loss of control over one’s eating is also part of the diagnostic criteria of binge eating disorder (BED) [4]. While a full manifestation of this disorder is rare in childhood and youth, affected children and adolescents often experience a marked feeling of LOC [5]. Overweight children with LOC were found to have a higher average weight and to suffer from more severe psychopathology than overweight children without LOC [6,7], and past studies have established LOC as a valid marker of disturbed eating behavior [5,8]. 

Models developed for adult BED patients stress a biased processing of food-related stimuli and inhibitory control deficits as important maintaining factors. While there is ample empirical evidence for these theories in adults with BED [9,10,11,12,13], there are only few studies with children and adolescents to date. These studies found some support for both processing biases [14,15] and inhibitory problems [16,17] in children and adolescents with eating pathologies and overweight/obesity.

Electroencephalographic (EEG) studies examined attentional biases to food cues by focusing on the P3, a positive deflection with a parieto-central distribution starting roughly 300 ms post-stimulus onset [18]. The P3 has been linked to frontal top-down attentional mechanisms as well as to temporo-parietal processes of updating and memory storage [19]. Increased P3 amplitudes could be found for task-relevant versus task-irrelevant stimuli, but also for appetitive compared to neutral stimuli [19,20,21]. In addition, P3 amplitudes were higher when target stimuli were emotionally significant than when neutral targets were presented, pointing to converging effects of attention and emotion on P3 amplitudes [22].

Previous studies investigating the processing of passively viewed food images found higher P3 amplitudes when participants viewed pictures of food versus neutral objects [23,24], were hungry versus satiated [25], or when the viewed high-calorie food was available versus unavailable [26]. In contrast, the impact of body weight on P3 amplitudes is less clear. Some studies found overweight/obese adolescents and adults not to differ from normal-weight controls when viewing food images [24,27]. However, when adolescents completed a Go/NoGo task, with the Go stimuli being food and the NoGo stimuli being non-food items, heavier children showed reduced P3 amplitude differences in Go versus NoGo stimuli, which was interpreted as reflecting reduced inhibitory functioning [21]. Reduced P3 amplitudes have furthermore been linked to heightened impulsivity [28,29], which studies found to be increased in children and adolescents with eating pathologies and overweight/obesity [16,17].

To our knowledge, no EEG study to date investigated the processing of visual food stimuli in children or in adolescents with LOC. Our study thus used a Go/NoGo task with visual food and non-food stimuli presented as task-irrelevant distractors and compared a sample of adolescents reporting LOC in the past month with a healthy control group. We hypothesized that both groups would show increased P3 amplitudes in Go compared to NoGo trials. Given the previously reported converging effects of task-relevant and emotionally salient stimuli [22], we expected amplitudes to be highest in Go trials with food distractors. However, we also expected adolescents with LOC to show reduced amplitude differences in food versus non-food distractors in both Go and NoGo trials, signifying reduced inhibitory control and increased impulsivity when being confronted with food images. In addition, LOC should be associated with increased snack food consumption, with participants with LOC consuming more of the subsequently offered snack food than controls.

## 2. Materials and Methods

Adolescents with at least one self-reported episode of loss of control eating in the last four weeks (LOC group) and a matched control group (CG) between the ages of 9 and 16 years were recruited via newspaper articles, e-mail announcements, and brochures handed to local pediatric and general practitioner practices. After a diagnostic screening by phone, potential participants and a parent or guardian were invited for an in-depth diagnostic assessment. Adolescents were excluded from participation if they reported a history of seizures, compensatory weight control behavior, psychotic symptoms, suicidality, medical conditions influencing weight and eating behavior, food allergies, intolerances, or self-imposed dietary restrictions. Both the adolescent and the parent or guardian gave their informed consent after the study was explained to them. Participants were informed that the exact hypotheses guiding the study could only be revealed afterwards to avoid biasing the results. All study procedures were in accordance with the Declaration of Helsinki [30] and approved by the ethics committee of the medical faculty (project number 557/2015B01).

Power calculations based on previous research with adults [31] yielded minimum sample sizes of 13–16 participants per group, so a total of 40 adolescents were recruited for this study. The Diagnostic Interview for Mental Disorders in Children and Adolescents (Kinder-DIPS) [32] was used to determine the presence of a mental disorder. Eating pathology was additionally assessed by means of the Child Eating Disorder Examination-Questionnaire (ChEDE-Q) [33], and the short version of the Food Craving Questionnaire—Trait (FCQ-T-reduced) [34]. All participants were also measured and weighed in light clothing. EEG appointments were scheduled on a different day, usually in the afternoon and within two weeks of the diagnostic assessment. All participants were offered an ad libitum meal before the EEG appointment to standardize hunger levels. Six data sets (one LOC) were lost because of excessive noise or movement artifacts and technical problems, respectively. The final sample in the data analysis thus consisted of 34 adolescents, 15 in the LOC group, and 19 in the control group. The two groups were not significantly different in gender distribution (*p* = 0.510), mean age (*p* = 0.419), and mean BMI percentile [35] (*p* = 0.193); however, the LOC group reported significantly more eating pathology on both the ChEDE-Q (*p* = 0.003) and the FCQ-T-r (*p* = 0.002) (see Table 1 for means (Ms) and standard deviations (SDs)).

Participants completed a color-based Go/NoGo task, with a numerical target or non-target stimulus (Numbers 1–4) being presented on one side of the screen and an irrelevant high-calorie food or neutral stimulus being presented on the opposite side (112 trials per condition; see Figure 1). Pictures were purchased from a commercial media platform (Shutterstock, Inc., New York, NY, USA) to ensure maximum comparability of high-calorie food and neutral pictures. Luminance of all picture pairs and the numerical stimuli was adjusted using a modified script from the SHINE toolbox in Matlab (The MathWorks, Natick, MA, USA) [36]. Picture size was 500 by 300 pixels. All pictures were displayed on a neutral grey background for 300 ms with the interstimulus interval (ISI) jittered to last between 2 and 2.2 s. Each picture was presented twice per Go condition and twice per NoGo condition, once to the left of the fixation cross and once to the right. Pictures were randomized and paired with a target or a non-target stimulus (defined by their color), leading to an equal number of Go trials and of NoGo trials. In each trial, the identity of the number was randomly selected to be equally likely any of the numbers from 1 to 4.

Participants were seated in a sound-attenuated booth with controlled lighting, 60 cm from the screen, with both of their index and middle fingers resting on an RB-840 response pad (Cedrus Corporation, San Pedro, CA, USA). They were instructed to press the response key assigned to the numerical target stimulus (numbers from 1 to 4) and to withhold their response if no target stimulus was presented. The color of the target (Go) stimuli was selected to be either turquoise or purple, with the other color being used for the non-target (NoGo) stimuli. This was counterbalanced across participants. After participants completed 12 practice trials with animal pictures as distractor stimuli, the study assistant left the booth and the main experiment was started.

After the EEG experiment, the elastic cap holding the electrodes was removed and participants were led to another room with a snack buffet consisting of 10 different snack foods with a total caloric value of approximately 14,600 kcal. Participants were told that the study was looking at which foods they liked best and that they could eat as much as they liked. They were asked to try every item at least once and fill in a rating of preference for the different snacks. Participants were then left alone with the snack buffet for 20 min, after which the study assistant re-entered the room. All snacks were weighed before and afterwards to calculate the exact amount of consumed macronutrients. At the end of the experiment, participants rated the previously viewed stimuli in randomized order for valence and arousal on a SAM scale [37]. In addition, food pictures were rated for palatability on a continuous scale ranging from 1 (lowest rating) to 9 (highest rating) [38]. The duration of the entire experiment including preparation was ~2.5 h.

EEG data were collected from 64 Ag/AgCl active electrodes placed according to the extended 10–20 system using actiCap (Brain Products GmbH, Gilching, Germany). Data were recorded in relation to a reference electrode placed on the left earlobe and re-referenced offline to linked earlobes, with a sampling rate of 500 Hz. Four additional passive electrodes were placed above and below the right eye as well as on both outer canthi to monitor eye movement. Data were analyzed using BrainVision Analyzer 2.1 (Brain Products GmbH): Data were filtered with a 50 Hz notch filter to remove power line noise and a band-pass filter of 0.1–30 Hz. After manual inspection of the raw data for noise and artifacts, eye movement artifacts were corrected [39], and the data were segmented into 1.2 s epochs around picture onset (200 ms pre-stimulus) and baseline-corrected (−200 ms to stimulus onset). Based on the literature [21,40] and grand average topography across all conditions and participants, channels Oz, O1, O2, POz, PO3, and PO4 in the time window including peak activity (300–360 ms post-stimulus onset) were selected for P3 analysis.

Mean activity in the selected channels for these time windows was exported for each participant and further analyzed using IBM^®^ SPSS^®^ Statistics 25 (IBM Corp., Armonk, NY, USA). Mean amplitudes were entered into mixed model analyses of variance (ANOVAs) with the between-subjects variable Group (LOC and CG) and the within-subject variable Go/NoGo (Target present versus Target absent) and Distractor category (Food versus Neutral). Mean reaction times and error rates were entered into mixed model ANOVAs with the between-subjects variable Group (LOC and CG) and the within-subject variable Distractor category (Food versus Neutral). Mean valence and arousal ratings were entered into mixed model ANOVAs with the between-subjects variable Group (LOC and CG) and the within-subject variable Distractor category (Food vs. Neutral). Mean palatability ratings were analyzed using an independent samples *t*-test. Values of *p* ≤ 0.05 were considered significant.

## 3. Results

### 3.1. Behavioral Data and Stimulus Ratings

There was no group difference in consumption of the lye pretzel in the ad libitum meal, *t*_(32)_ = 0.003, *p* = 0.998, with both groups consuming a mean of 33 g of the pretzel. Groups did also not differ in mean desire to eat reported before the start of the EEG experiment (LOC group: mean = 2.4, control group: mean = 3.0 on a continuous 9-point scale), *t*_(32)_ = 1.06, *p* = 0.299.

There was no significant difference in calorie consumption between the two groups, *t*_(32)_ = 1.17, *p* = 0.251, with both groups consuming a substantial amount of the offered high-calorie food (see Table 1). For macronutrient consumption, there was a trend-level difference, with the LOC group tending to consume more protein than the control group, *t*_(32)_ = 1.71, *p* = 0.097. Carbohydrate consumption (*p* = 0.459) and fat consumption (*p* = 0.157) were not significantly different between the two groups.

For mean reaction times, there was no significant main effect of distractor category, *F*_(1,32)_ = 1.32, *p* = 0.259 (see Table 1 for Ms and SDs of all behavioral data), or group, *F*_(1,32)_ = 0.67, *p* = 0.418, and no significant interaction, *F*_(1,32)_ = 2.71, *p* = 0.110. For mean percentage of incorrect responses, there was also no significant main effect of distractor category, *F*_(1,32)_ = 0.64, *p* = 0.429, or group, *F*_(1,32)_ = 0.23, *p* = 0.636, and no significant interaction, *F*_(1,32)_ = 1.68, *p* = 0.204. The same was true for mean misses with no effect of distractor category, *F*_(1,32)_ = 0.42, *p* = 0.632, or group, *F*_(1,32)_ < 0.01, *p* = 0.996, and no interaction, *F*_(1,32)_ = 0.10, *p* = 0.877, and for number of false alarms with no effect of distractor category, *F*_(1,32)_ = 1.04, *p* = 0.316, or group, *F*_(1,32)_ = 0.89, *p* = 0.353, and no interaction, *F*_(1,32)_ < 0.01, *p* = 0.979.

For mean valence ratings, there was a significant main effect of distractor category, *F*_(1,32)_ = 5.58, *p* = 0.024 (see Table 1 for Ms and SDs of all rating data), with high-calorie food stimuli being rated as significantly more positive than neutral stimuli. There was no significant main effect of group, *F*_(1,32)_ = 1.79, *p* = 0.191, and no significant interaction, *F*_(1,32)_ = 0.54, *p* = 0.466. For mean arousal ratings, there was a marginally significant main effect of distractor category, *F*_(1,32)_ = 4.17, *p* = 0.050, with high-calorie food stimuli being rated as significantly more arousing than neutral stimuli. There was no significant main effect of group, *F*_(1,32)_ = 0.25, *p* = 0.618, and no significant interaction, *F*_(1,32)_ = 0.001, *p* = 0.982. Furthermore, there was no significant difference in mean palatability ratings for the high-calorie food stimuli between the two groups, *t*_(32)_ = 0.11, *p* = 0.912.

### 3.2. Electroencephalographic Data

Mean P3 amplitudes for all trials showed a significant main effect of Go/NoGo, *F*_(1,32)_ = 104.64, *p* < 0.001, with significantly higher mean amplitudes in Go trials than for NoGo trials (22.0 µV and 15.6 µV), and a significant main effect of the Distractor category, *F*_(1,32)_ = 18.36, *p* < 0.001, with significantly higher mean amplitudes in food distractors versus neutral distractors (19.5 µV and 18.1 µV). In addition, there was a significant three-way interaction of Go/NoGo, Distractor category, and Group, *F*_(1,32)_ = 4.21, *p* = 0.048 (see Figure 2 for grand average time courses and topographies, and Table 2 for Ms and SDs). There were no further significant main effects or interactions.

Post-hoc *t*-tests conducted separately for Group showed that in the Go trials, the LOC group’s mean P3 amplitudes were comparable irrespective of distractor category, *t*_(14)_ = 1.52, *p* = 0.152. In NoGo trials, however, mean P3 amplitudes were significantly higher when the distractor was a high-calorie food stimulus than when it was a neutral stimulus, *t*_(14)_ = 3.89, *p* = 0.002. In contrast, the control group’s mean P3 amplitudes were significantly higher when a Go stimulus was paired with a high-calorie food stimulus compared to a neutral stimulus, *t*_(18)_ = 5.30, *p* < 0.001, but not when a NoGo stimulus was paired with these distractors, *t*_(18)_ = 1.15, *p* = 0.266. Independent samples *t*-tests showed no significant between-group differences (all *p* > 0.4). To rule out topographical between-group differences as the underlying cause for these findings, independent samples *t*-tests were conducted for P3 amplitudes in more anterior electrodes (POz, PO3, and PO4) versus posterior electrodes (Oz, O1, and O2) in Go and NoGo trials and in food and neutral stimuli, respectively. These tests showed no significant between-group differences (all *p* > 0.4).

## 4. Discussion

This study strived to examine the processing of distracting visual food stimuli in adolescents with LOC and a healthy control group. As hypothesized, both groups showed increased P3 amplitudes in Go compared to NoGo trials, which is in line with previous findings of higher P3 amplitudes for task-relevant target stimuli [19]. In addition, P3 amplitudes were increased when food distractors compared to non-food distractors were presented. This corresponds to previous reports of increased amplitudes for salient stimuli in general, and for pictures of high-calorie food in particular [23,24]. In line with our hypotheses, adolescents with LOC furthermore showed reduced amplitude differences in food versus non-food distractors in Go trials compared to the healthy control group. Contrary to our expectations, however, this pattern was reversed in the NoGo trials. In these trials, the LOC group showed significant amplitude differences between food and non-food distractors, while the control group did not. One way to interpret this pattern of results is by means of Gray’s reinforcement sensitivity theory [41]. Briefly, this theory assumes that the behavioral approach system (BAS) mediates the behavioral approach with a stimuli signaling reward, while the behavioral inhibition system (BIS) mediates behavioral inhibition with a stimuli signaling punishment, and sensitivity differences of these systems are thought to account for interindividual differences in approach/avoidance behavior [42,43].

Applying the BIS/BAS approach to our findings, the presented results could point to the possibility of distracting visual food stimuli activating different neurophysiological systems in adolescents with LOC versus healthy control adolescents, while Go/NoGo stimuli activate identical systems. Given previous unpleasant experiences of loss of control during eating as well as negative mealtime family interaction [44] in the LOC group, visual food stimuli might come to be associated with negative emotional states, thus taking the form of conditioned aversive stimuli and activating the BIS. This hypothesis is supported by a negative association of LOC frequency and valence ratings for the high-calorie stimuli in our sample—the more frequently adolescents experienced LOC episodes over the last four weeks, the more negatively they rated the pictures of high-calorie food. In contrast, no such association is likely to be present in the healthy control group. For these adolescents, the consumption of food is always a pleasant experience. Visual food stimuli might thus take the form of conditioned appetitive stimuli, activating the BAS. It is reasonable to assume BAS activation in Go trials for all participants, and accordingly, previous research found larger P3 Go amplitudes in participants with a high versus low trait BAS [29,45].

In our study, the LOC group’s reduced difference amplitudes in Go trials with food distractors could therefore be explained by increased BIS activation in response to visual food stimuli, “cancelling out” at least part of the BAS activation caused by the Go trial. The opposite would be true for NoGo trials. In these trials, BIS activation in response to visual food stimuli and increased inhibitory activation caused by the NoGo trial should show converging effects, leading to higher difference amplitudes in the group with LOC. Importantly, this pattern should be reversed in the healthy control group, with converging BAS effects in Go trials with food distractors leading to increased difference amplitudes in the Go condition. This is exactly what we found in our analyses. Further support for this interpretation comes from a study using a Go/NoGo task and reporting reduced P3 amplitude differences in food (Go) versus non-food (NoGo) stimuli in heavier adolescents [21], and from a study reporting reduced oddball P3 amplitudes to visual food stimuli in obese adults [46]. In addition, activity in the P3 time window was found to be sensitive to the fat content of visual food stimuli and might thus be particularly relevant for later-stage categorization and decision-making regarding high-calorie food stimuli [47].

Importantly, this study has several limitations beyond the small sample size. First, adolescents needed to report at least one incident of LOC in the last four weeks to be included in the LOC group. While this is comparable to previous studies investigating LOC [48,49,50], it does not satisfy the proposed research criteria for LOC disorder [5,51], which require more frequent episodes over a longer period of time. It would furthermore be advisable to screen participants for use of cannabinoids to avoid possible misclassifications. However, the increased eating pathology and trait food craving of the LOC group still indicate the validity of our classification. In addition, the macronutrient composition of the ad libitum meal is not known. Future studies might benefit from offering standardized food items allowing for the calculation of pre-experimental macronutrient consumption as an additional variable. Given the presumed association of BIS activation to visual food stimuli and unpleasant food-related experiences caused by LOC eating and/or negative mealtime family interaction, this aspect of our findings should be investigated further. Especially ecological momentary assessment studies should take care to collect ratings of negative feelings during LOC episodes and during family mealtime in general, as these might be important predictors of aberrant food stimuli processing.

Another potential limitation concerns the design of the experimental task. In the current version of this task, Go trials and NoGo trials are presented with the same frequency. Although a similar task design was used in a previous study with overweight children [52], the Go condition in our study was not established as the prepotent response. Consequently, the amount of inhibition required in this study might have been less extensive than in conventional Go/NoGo paradigms, thereby reducing the size of the reported effects. While the present task thus allows for the investigation of food picture processing that is not directly relevant to the experimental task, it might benefit from modification to present NoGo stimuli less frequently. This could be done in a task similar to the one used by Watson and colleagues [53] and it could increase the power to detect modulations of P3 amplitudes [54]. As the interpretation of our findings along the lines of BIS/BAS is rather speculative at the moment, it would moreover be advisable to collect self-report data on participants’ BIS/BAS sensitivity [55]. This would allow more in-depth analyses and a better interpretation of the obtained results.

## 5. Conclusions

To conclude, this is the first study investigating EEG correlates of visual food stimulus processing in an adolescent sample with LOC and a healthy control group of comparable weight. The results indicate that distracting visual food stimuli might activate different neurophysiological systems in adolescents with LOC from those in healthy control adolescents. Specifically, visual food stimuli might have come to be associated with negative emotional states in adolescents with LOC, thus taking the form of conditioned aversive stimuli and activating the BIS, whereas the BAS would be activated in adolescents without eating pathologies. As such, the results provide new insights into the altered inhibitory functioning of adolescents with eating pathologies, which can develop into a full-fledged binge eating disorder if left untreated [56]. Future studies should thus focus on further investigating inhibitory functions in relation to BIS and BAS, as this might prove a worthwhile approach to a better understanding of the underlying processes in LOC.

## Figures and Tables

**Figure 1 nutrients-11-00210-f001:**
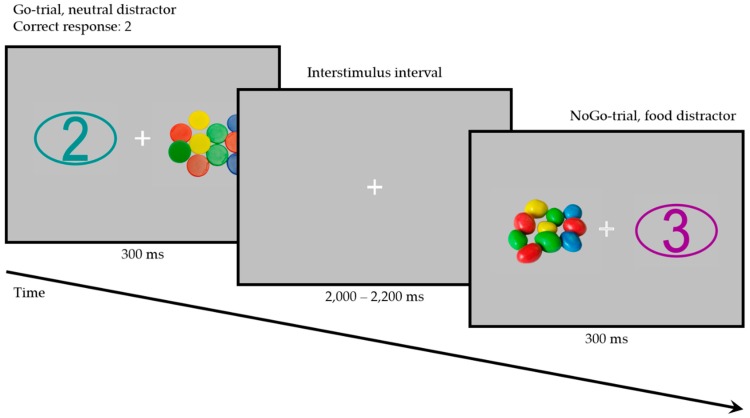
Example of a Go trial with a neutral distractor and Target Number 2 (left; target color: turquoise) and of a NoGo trial with a high-calorie food distractor (right). Note: The distractor stimuli depicted here are for illustrative purposes only and were not used in the study.

**Figure 2 nutrients-11-00210-f002:**
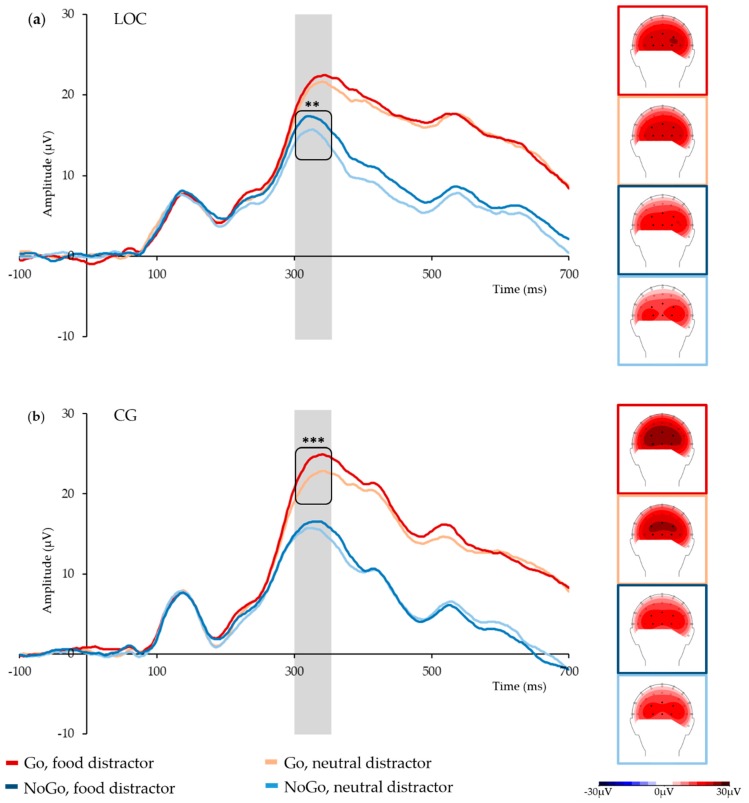
Grand average P300 time courses in the analyzed electrode cluster, and topographies by condition and group: (**a**) group with loss of control eating (LOC); (**b**) control group (CG). The P300 time window is marked in grey; black dots mark the analyzed electrodes; ** marks *p* < 0.01, *** marks *p* < 0.001.

**Table 1 nutrients-11-00210-t001:** Descriptive and behavioral data for the group with loss of control eating (LOC) and the matched control group (CG).

Descriptive and Behavioral Data	LOC	CG
*n* (girls)	15 (9)	19 (9)
Age	12.9	12.3
BMI percentile	75.8 (29.8)	60.5 (35.9)
ChEDE-Q	1.8 (1.3) **	0.5 (0.6) **
FCQ-T-r	35.2 (13.2) **	21.9 (5.1) **
Energy intake		
Total intake (kcal)	661 (264)	558 (249)
Fat (%)	18.2 (3.8)	16.7 (4.7)
Carbohydrates (%)	49.9 (4.5)	51.2 (3.1)
Protein (%)	6.3 (1.2) ^#^	5.5 (1.2) ^#^
Reaction time (ms)		
High-calorie food distractor	733 (106)	784 (152)
Neutral distractor	754 (133)	780 (144)
% Incorrect responses		
High-calorie food distractor	5.1 (6.4)	5.5 (8.7)
Neutral distractor	3.7 (6.2)	5.8 (8.6)
% Misses		
High-calorie food distractor	0.06 (0.09)	0.06 (0.11)
Neutral distractor	0.06 (0.09)	0.06 (0.13)
False alarms		
High-calorie food distractor	0.3 (0.7)	0.7 (1.2)
Neutral distractor	0.5 (0.9)	0.9 (1.9)
Valence		
High-calorie food *	6.3 (1.3)	6.0 (1.2)
Neutral *	5.9 (1.7)	5.2 (1.1)
Arousal		
High-calorie food *	3.1 (1.9)	3.4 (1.9)
Neutral *	2.6 (1.5)	2.9 (1.6)
Palatability	5.7 (1.5)	5.8 (1.2)

Means and standard deviations (SDs) (in parentheses) for BMI percentile, eating pathology on the Child Eating Disorder Examination Questionnaire (ChEDE-Q; mean global score) and the short version of the Food Craving Questionnaire Trait (FCQ-T-r; sum score); energy intake and relative macronutrient percentage; valence, arousal, and palatability ratings; reaction times and error rate, by group and condition (where applicable). ^#^ marks differences ≤0.1; * marks significant differences ≤0.05, ** marks significant differences ≤0.01.

**Table 2 nutrients-11-00210-t002:** Mean EEG amplitudes for the group with loss of control eating (LOC) and the matched control group (CG).

Condition	LOC	CG
Go, food distractor	21.3 (12.7)	23.9 (7.0) ***
Go, neutral distractor	20.4 (12.3)	21.9 (7.1) ***
NoGo, food distractor	16.6 (11.4) **	16.0 (8.0)
NoGo, neutral distractor	14.8 (11.3) **	15.1 (6.5)

Means and SDs (in parentheses) for P300 amplitudes by group and condition. Asterisks mark significant differences between food and neutral distractor, separately for group and Go vs. NoGo stimulus. ** marks significant differences ≤0.01; *** marks significant differences ≤0.001.

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
