# Peer review of "Altered Processing of Visual Food Stimuli in Adolescents with Loss of Control Eating"

_nutrients, 2019, doi:10.3390/nu11020210_

Reviewer 1 Report

This study has the aim to investigate the ERP modulation in LOC subjects. According to my opinion, it's not correct to speak of 'altered processing of food stimuli'. Please, explain what kind of processing you are exploring (and review it in the title and in the text). Surely it is not a chemosensory processing (i.e, linked to food), It could be related to a visual recognition of food or an emotional processing related to food recognition in LOC subjects. Please, correct the form 'mean P300 amplitude', you can use simply 'P3 amplitude, as suggested in Luck (2005).

Please, review, in a more accurate way, the conclusion the text

Author Response

Response to Reviewer 1 – Comments

 Point 1: This study has the aim to investigate the ERP modulation in LOC subjects. According to my opinion, it's not correct to speak of 'altered processing of food stimuli'. Please, explain what kind of processing you are exploring (and review it in the title and in the text). Surely it is not a chemosensory processing (i.e, linked to food), It could be related to a visual recognition of food or an emotional processing related to food recognition in LOC subjects.

Response 1: We would like to thank the reviewer for drawing our attention to this lack of clarity. As suggested, we now refer to visual food stimuli in the title and throughout the text.

Point 2: Please, correct the form 'mean P300 amplitude', you can use simply 'P3 amplitude, as suggested in Luck (2005).

Response 2: We have changed this term as suggested.

Point 3: Please, review, in a more accurate way, the conclusion the text

Response 3: We have revised the conclusion to provide a more detailed summary of the obtained results.

Reviewer 2 Report

The authors recruited two groups of adolescents with or without at least one episode of Loss of control eating. They used EEG and a Go/NoGo task with Food and non-Food distractors in order to investigate group differences in P300. They found a 3-way interaction between task condition, group and distractor category. Results are interpreted in the light of Gray’s reinforcement sensitivity theory.  

The manuscript is well written, carefully phrased, and concise. However, the unclear task design, relatively small sample size, categorisation of the study groups (i.e. prerequisite of only one episode of LOC), and, so far, rather speculative discussion limit the generalisability of the results and, consequently, my enthusiasm for this study.

Specific comments:

Introduction: 

I could not entirely follow the authors’ reasoning that lead them to some of the expectations. In particular, it would be worthwhile to elaborate a bit on why exactly the authors expected “adolescents with LOC to show reduced amplitude differences for food versus non-food distractors on both Go and NoGo trials, signifying reduced inhibitory control when being confronted with food images.” Being no EEG expert, I had the impression that the information given in the introduction did not highlight that the P300 is associated with inhibitory control. 

In the introduction, the authors take the effort to distinguish between the P3a and P3b – a distinction that they never get back to. This could be skipped for the sake of streamlining the MS.

Additional work from the groups of U Toepel and K Ohla might also be relevant to the current study. 

Material and methods: 

Given that the inclusion criteria assessed whether adolescents experienced at least one episode of LOC during the past 4 weeks and adolescence being a period for higher likelihood of consumption of drugs that might lead to the same experience: Where participants screened/asked for drug use, especially cannabinoids?

Establishing a standardised hunger level for the main experiment was a good idea. Did the authors quantify how much of each macronutrient participants consumed at the ad lib meal, too? I think this might be interesting for the EEG data, and also in order to interpret the snack intake data. Did they measure subjective hunger levels before / after the meal and EEG measurement?

ll 94-96: Excluded data sets do inversely match the group sizes of the final sample, I guess something got confused here. 

The description of the task lacks detail: 

Please clarify what the term “condition” refers to: Is it distractor category or go / NoGo condition? 

Because this was not always clear to me, I might have missed relevant information. 

However, I would like to encourage the authors to explicitly include the following information: 

What was the ratio of Go vs. NoGo trials? It is common practice to make Go trials much more frequent than NoGo trials in order to establish a prepotent response that makes inhibitory control of an already prepared response necessary. If the ratio was one, the authors might have measured something different from inhibition normally seen in Go/NoGo tasks. Please include this information. 

What was the timing of the experimental day? Did participants arrive over the whole day/always in the morning/ did time for snack buffet differ between groups? How long took the EEG experiment including preparation?

Please include actual stimulus material in Figure 1. How were food stimuli selected? Is there information on macronutrients, and the perceived potential to trigger loss of control? 

Where there any time constraints on the response window? How were errors defined, i.e. only commission errors or also misses?

Was there any preprocessing of RTs and all other data such as outlier analysis, transformations, tests of assumptions for statistical tests such as normality (of residuals)?

Which items were offered on the snack buffet? Where adolescents with vegan/vegetarian lifestyle excluded? Which categories were included in the picture material?

The correlation analysis is not mentioned in the methods.

The sample covered a relatively large age range during which major developmental changes, especially in inhibitory capacity, are to be expected. Were any of the analyses corrected for age? Did the authors (exploratively, I do acknowledge the small sample size and its limitations) check for an effect of sex? 

Results: 

Please give exact p-values. Interpreting p=.097 as trend-level difference and referring to ps>=.1 as no significant difference seems a bit awkward. 

Layout of the Tables is not intuitive. I would like to advise the authors to emphasise a bit more what entries in column 1 are representing a subheading and which not. 

I would find the presentation of the snack results more informative if (a), as is done already, energy intake is presented in kcal, and (b) in addition, the percentage of each macronutrient is given. As it is presented now the reader has to muse whether marginal differences can be explained by overall intake difference or differences in macronutrient preference. 

Discussion: 

Although largely speculative, the authors make an interesting point stating that their data could be explained by Gray’s reinforcement sensitivity theory. Can the authors think of a way to substantiate this a bit more with their data, e.g. shouldn’t some of the conditioned aversiveness/appetitiveness be reflected in ratings the participants made? And, maybe more importantly, NoGo amplitude-differences were positively associated with intake – i.e. an effect presumably triggered by the BIS is in real life associated with an approach reaction, which I find counterintuitive. I would appreciate clarification. 

Further, can the authors think of a way to quantify severity of LOC and test the association between this measure and effect size? Or do the authors have information on variables that could represent variance in unpleasant experiences/ negative family mealtime interaction? 

Author Response

Response to Reviewer 2 – Comments

The authors recruited two groups of adolescents with or without at least one episode of Loss of control eating. They used EEG and a Go/NoGo task with Food and non-Food distractors in order to investigate group differences in P300. They found a 3-way interaction between task condition, group and distractor category. Results are interpreted in the light of Gray’s reinforcement sensitivity theory.  

The manuscript is well written, carefully phrased, and concise. However, the unclear task design, relatively small sample size, categorisation of the study groups (i.e. prerequisite of only one episode of LOC), and, so far, rather speculative discussion limit the generalisability of the results and, consequently, my enthusiasm for this study.

We would like to thank the reviewer for the positive assessment of our manuscript. We hope that our revision addresses the concerns mentioned above.

Specific comments: 

Introduction:

Point 1: I could not entirely follow the authors’ reasoning that lead them to some of the expectations. In particular, it would be worthwhile to elaborate a bit on why exactly the authors expected “adolescents with LOC to show reduced amplitude differences for food versus non-food distractors on both Go and NoGo trials, signifying reduced inhibitory control when being confronted with food images.” Being no EEG expert, I had the impression that the information given in the introduction did not highlight that the P300 is associated with inhibitory control.

Response 1: We agree that this aspect was not described properly in the introduction. We have therefore provided more background on the link between reduced P3 amplitudes and increased impulsivity and on how this relates to the investigation of overweight/obesity.

Point 2: In the introduction, the authors take the effort to distinguish between the P3a and P3b – a distinction that they never get back to. This could be skipped for the sake of streamlining the MS.

Response 2: We agree that this distinction is not necessary for understanding our study; we have modified the paragraph accordingly.

Point 3: Additional work from the groups of U Toepel and K Ohla might also be relevant to the current study.

Response 3: We would like to thank the reviewer for this suggestion. We have included some of this work in the introduction and discussion sections.

Material and methods: 

Point 4: Given that the inclusion criteria assessed whether adolescents experienced at least one episode of LOC during the past 4 weeks and adolescence being a period for higher likelihood of consumption of drugs that might lead to the same experience: Where participants screened/asked for drug use, especially cannabinoids?

Response 4: This is a valid point. Unfortunately, participants were not screened for use of cannabinoids, as this is not commonly done in studies of loss of control eating. However, we realize that this might lead to possible misclassifications of participants and it is now included as a limitation.

Point 5: Establishing a standardised hunger level for the main experiment was a good idea. Did the authors quantify how much of each macronutrient participants consumed at the ad lib meal, too? I think this might be interesting for the EEG data, and also in order to interpret the snack intake data. Did they measure subjective hunger levels before / after the meal and EEG measurement?

Response 5: This is an important point. Since the ad libitum meal consisted of a locally bought lye pretzel, we are unfortunately unable to provide the macronutrient content. This is now included as a limitation. However, we recorded each pretzel’s initial weight as well as the weight of any leftovers: Children consumed an average of 33 g of pretzel with no difference between the two groups (= .998). We also measured subjective desire to eat before the start of the EEG measurement, and this was not significantly different between the two groups (p = .299). This information is now included in the manuscript.

Point 6: ll 94-96: Excluded data sets do inversely match the group sizes of the final sample, I guess something got confused here.

Response 6: We have clarified how many data sets were entered into the final analysis.

Point 7: The description of the task lacks detail: 

Please clarify what the term “condition” refers to: Is it distractor category or go / NoGo condition?

Response 7: We would like to thank the reviewer for pointing out this lack of clarity. We have added the missing information.

Point 8: Because this was not always clear to me, I might have missed relevant information. 

However, I would like to encourage the authors to explicitly include the following information: What was the ratio of Go vs. NoGo trials? It is common practice to make Go trials much more frequent than NoGo trials in order to establish a prepotent response that makes inhibitory control of an already prepared response necessary. If the ratio was one, the authors might have measured something different from inhibition normally seen in Go/NoGo tasks. Please include this information.

Response 8: In our experimental task, the ratio of Go-trials and NoGo-trials was indeed one. This is now explicitly stated in the task description. Suggested changes to the experimental task were already briefly mentioned in the discussion and we have now further elaborated on that point.

Point 9: What was the timing of the experimental day? Did participants arrive over the whole day/always in the morning/ did time for snack buffet differ between groups? How long took the EEG experiment including preparation?

Response 9: Participants usually arrived in the afternoon; the duration of the entire EEG experiment including preparation was about 2.5 hours. Time for the snack buffet was standardized to be 20 min for all participants. We have added this information to the methods section.

Point 10: Please include actual stimulus material in Figure 1. How were food stimuli selected? Is there information on macronutrients, and the perceived potential to trigger loss of control?

Response 10: Since the visual stimuli were purchased from a commercial media platform, we unfortunately do not own the rights to have them printed in the manuscript (please find two examples of the original stimulus pairs below).

Food stimuli were selected to depict high-calorie food with high fat/carbohydrate content that is often eaten during food binges (based on clinical experience but see also Raymond et al., 2007). Unfortunately, we do not have detailed information regarding macronutrient composition of these foods or the perceived potential to trigger loss of control episodes.

Note: These stimuli must not be accessible online.

Point 11: Were there any time constraints on the response window? How were errors defined, i.e. only commission errors or also misses?

Response 11: Participants had to respond within 1.5 s of picture presentation. We defined errors as wrong responses (i.e. a number response that was different from the target number); this was clarified in the manuscript. As the analyses of misses and correct responses, respectively, produced very comparable results, we decided not to report this information to avoid cluttering the results section. However, we will be happy to add this information to the manuscript should the reviewer deem it necessary.

Point 12: Was there any preprocessing of RTs and all other data such as outlier analysis, transformations, tests of assumptions for statistical tests such as normality (of residuals)?

Response 12: One participant with LOC showed outlier values with regard to P300 amplitudes on distractor trials. However, as excluding this participant did not change the significance of the reported results, his data was included in the final analysis. The Kolmogorov-Smirnov test showed violations of the normality assumption for error rates as well as some of the rating data, but Mann-Whitney-U tests confirmed no significant group differences for these variables. Adjusted F- and p-values are reported if Levene’s test for equality of variances was significant. For the sake of readability we reported unadjusted degrees of freedom, but we will be happy to correct this if required.

Point 13: Which items were offered on the snack buffet? Where adolescents with vegan/vegetarian lifestyle excluded? Which categories were included in the picture material?

Response 13: The following items were offered on the snack buffet: Chocolate coated peanuts, pieces of chocolate, chocolate butter cookies, chocolate cream filled cookies, jelly candy, a baked savory snack, a fried savory snack, a savory corn and peanut snack, fried salt cookies, and savory roasted peanuts. The picture material consisted of pictures of sweet baked goods, cookies, chocolate, jelly candy, ice cream, chips, burgers, meatloaf, and deep-fried food. Adolescents with any allergies, intolerances, or self-imposed dietary restrictions were excluded from the study.

Point 14: The correlation analysis is not mentioned in the methods.

Response 14: Please see our response to Point 20. After careful consideration, we decided to remove the correlation analysis from the manuscript.

Point 15: The sample covered a relatively large age range during which major developmental changes, especially in inhibitory capacity, are to be expected. Were any of the analyses corrected for age? Did the authors (exploratively, I do acknowledge the small sample size and its limitations) check for an effect of sex?

Response 15: As we carefully matched our sample with regard to age, we did not correct for age in any of the analyses. We did not check for an effect of sex, but explorative analyses do not show any main effects or interactions for this additional variable.

Results: 

Point 16: Please give exact p-values. Interpreting p=.097 as trend-level difference and referring to ps>=.1 as no significant difference seems a bit awkward.

Response 16: We agree with the reviewer and now give the exact p-values.

Point 17: Layout of the Tables is not intuitive. I would like to advise the authors to emphasise a bit more what entries in column 1 are representing a subheading and which not.

Response 17: We agree with the reviewer that the current formatting of the tables is not ideal – this seems to have been changed in the submission process. We have reformatted the tables according to the reviewer’s suggestion.

Point 18: I would find the presentation of the snack results more informative if (a), as is done already, energy intake is presented in kcal, and (b) in addition, the percentage of each macronutrient is given. As it is presented now the reader has to muse whether marginal differences can be explained by overall intake difference or differences in macronutrient preference.

Response 18: We would like to thank the reviewer for this idea. We agree that relative macronutrient percentage is more informative and we have changed the table accordingly.

Discussion: 

Point 19: Although largely speculative, the authors make an interesting point stating that their data could be explained by Gray’s reinforcement sensitivity theory. Can the authors think of a way to substantiate this a bit more with their data, e.g. shouldn’t some of the conditioned aversiveness/appetitiveness be reflected in ratings the participants made?

Response 19: We would like to thank the reviewer for this very helpful and interesting suggestion. If previous unpleasant experiences of loss of control during eating do indeed cause visual food stimuli to become associated with negative emotional states (thus taking the form of conditioned aversive stimuli and activating the BIS), we would expect the amount of LOC episodes in the past four weeks to be negatively associated with the valence ratings mentioned above. If tested one-sided, this is actually substantiated by the data, with more LOC episodes being associated with lower valence ratings for high-calorie stimuli (r(13) = ‑.46, p=.04). While the generalizability of this finding is clearly limited given the small sample size, we still think it is worth mentioning in the manuscript; it is now included in the discussion section.

Point 20: And, maybe more importantly, NoGo amplitude-differences were positively associated with intake – i.e. an effect presumably triggered by the BIS is in real life associated with an approach reaction, which I find counterintuitive. I would appreciate clarification.

Response 20: We would like to thank the reviewer for pointing out this possible contradiction. We believe that this finding might relate to previous findings of dietary restraint (presumably controlled by BIS) being associated with increased snack consumption in stressful situations (see e.g. Roemmich et al., 2002 and Wardle et al., 2000). However, given that the correlations remained marginally significant after controlling for self-reported restraint, we would also need a measure of subjective stress to support our assumption – which we currently do not have. As this finding would thus clearly require to be discussed in more detail, but is not directly related to our experimental rationale, we decided to remove the correlations from the manuscript.

Point 21: Further, can the authors think of a way to quantify severity of LOC and test the association between this measure and effect size? Or do the authors have information on variables that could represent variance in unpleasant experiences/ negative family mealtime interaction?

Response 21: This is another very interesting suggestion. While conclusions in this case are unfortunately limited by the small sample size, there does seem to be an association between NoGo difference amplitudes for food minus neutral distractors and frequency of LOC episodes in the past four weeks (r(13) = .34) – but it fails to reach significance (p = .11). We have included this as a topic warranting further research in the discussion section.

Round  2

Reviewer 2 Report

I would like to thank the authors for their elaborate responses. All of my questions have been answered to my satisfaction. 

I would like to encourage the authors to add detailed information on commission errors and misses to the MS. Further, the MS should state explicitly that adolescents with allergies etc. have been excluded from participation in the study.  

Author Response

I would like to thank the authors for their elaborate responses. All of my questions have been answered to my satisfaction.

We would like to thank the reviewer for this positive assessment of our revision.

I would like to encourage the authors to add detailed information on commission errors and misses to the MS. Further, the MS should state explicitly that adolescents with allergies etc. have been excluded from participation in the study.

We have added information on commission errors and misses to the results section and to Table 1. Exclusion criteria with regard to food allergies, intolerances, and self-imposed dietary restrictions have now been added to the sample description in the Material and Methods section.